# Initial evaluation of nighttime restlessness in a naturally occurring canine model of osteoarthritis pain

David Knazovicky[1], Andrea Tomas[1], Alison Motsinger-Reif[2,3] and B. Duncan X. Lascelles[1,2,4]

[1] Comparative Pain Research Laboratory, Department of Clinical Sciences, College of Veterinary Medicine, North Carolina State University, Raleigh, NC, USA
[2] Center for Comparative Medicine and Translational Research, Department of Clinical Sciences, College of Veterinary Medicine, North Carolina State University, Raleigh, NC, USA
[3] NCSU Bioinformatics Research Center, Department of Statistics, North Carolina State University, Raleigh, NC, USA
[4] Center for Pain Research and Innovation, UNC School of Dentistry, Chapel Hill, NC, USA

## ABSTRACT

Chronic pain due to osteoarthritis (OA) can lead to significant disruption of sleep and increased restlessness. Our objective was to assess whether naturally occurring canine OA is associated with nighttime restlessness and so has potential as a model of OA-associated sleep disturbance. The study was designed as a two-part prospective masked, placebo-controlled study using client-owned dogs (Part A $n = 60$; Part B $n = 19$). Inclusion criteria consisted of OA-associated joint pain and mobility impairment. The primary outcome measure for both parts was nighttime accelerometry. In Part B, quality of sleep was assessed using a clinical metrology instrument (Sleep and Night Time Restlessness Evaluation Score, SNoRE). Part A included dogs receiving two weeks of non-steroidal anti-inflammatory drug (NSAID) preceded with two weeks of no treatment. Part B was a crossover study, with NSAID/placebo administered for two weeks followed by a washout period of one week and another two weeks of NSAID/placebo. Repeated measures analysis of variance was used to assess differences between baseline and treatment. There were no significant changes in accelerometry-measured nighttime activity as a result of NSAID administration. SNoRE measures indicated significant improvements in aspects of the quality of nighttime sleep that did not involve obvious movement. These results reflect the few similar studies in human OA patients. Although accelerometry does not appear to be useful, this model has potential to model the human pain-related nighttime sleep disturbance, and other outcome measures should be explored in this model.

## INTRODUCTION

The annual economic cost of chronic pain in the United States is estimated to be $560–635 billion, with arthritis accounting for approximately $190 billion (1/3rd) of that total (*Medicine, 2011*). The same report recommended various strategies to help address this

Corresponding author
B. Duncan X. Lascelles,
Duncan_Lascelles@ncsu.edu

problem, and these included the development of more and better animal models of chronic pain. Our report describes the initial evaluation of a naturally occurring animal model of arthritis as a model of arthritis-induced sleep disturbance.

One-third to half of those people with painful arthritis suffer from sleep disturbances (*Moldofsky, Lue & Saskin, 1987*; *Power, Perruccio & Badley, 2005*; *Wilcox et al., 2000*; *Woolhead et al., 2010*; *Taylor-Gjevre et al., 2011*) often associated with restlessness (*Leigh et al., 1988*). Pain is thought to mediate a substantial amount of the relationship between arthritis and sleep problems (*Power, Perruccio & Badley, 2005*; *Woolhead et al., 2010*). Resulting sleep deprivation has been shown to have hyperalgesic effects, worsening the overall pain state (*Onen et al., 2001*; *Kundermann et al., 2004*). A recent focus group study highlighted the importance of night pain in human osteoarthritis (OA), and the relative lack of studies on night pain (*Woolhead et al., 2010*).

In dogs, OA is a common condition affecting a large percentage of the population. The disease process of canine OA of the hip is considered to be very similar to human OA (*Clements et al., 2006*) and therefore a potentially useful model—a spontaneous disease model (*Innes & Clegg, 2010*). Additionally, such a model has the added benefit that these companion dogs share the same environment as humans, making the model more relevant.

Recently, the use of activity monitors as an objective measurement of a dog's spontaneous activity has been described and validated (*Hansen et al., 2007*; *Brown, Boston & Farrar, 2010*; *Michel & Brown, 2011*). Day-time activity, as measured by an accelerometer, increases when non-steroidal anti-inflammatory (NSAID) analgesia is administered (*Brown, Boston & Farrar, 2010*; *Wernham et al., 2011*) or when therapeutic diets are introduced (*Rialland et al., 2012*).

Our clinical veterinary experience is that oftentimes, owners of dogs with OA report that their pets suffer nighttime restlessness (unpublished observations); clinical observations suggest this dissipates when analgesics (such as NSAIDs) are administered. This nighttime restlessness may be a manifestation of pain-associated sleep disturbance as seen in human OA patients, and so could present a good pre-clinical model of this complex state associated with OA. If day-time activity can be measured using accelerometry, and is measurably increased when analgesics are administered to dogs with OA-pain, it is possible that any nighttime restlessness associated with pain may be measureable in the same population.

The aim of our study was to perform preliminary investigations into whether naturally occurring canine OA is associated with nighttime restlessness as assessed using accelerometry and a clinical metrology instrument, and whether this might be a potential model of OA-associated sleep disturbance in humans.

We hypothesized that NSAID-responsive nighttime restlessness occurs in dogs suffering from painful OA. This study used objective accelerometry (AM) and a novel clinical metrology instrument (Sleep and Night Time Restlessness Evaluation Score; SNoRE) as primary outcome measures to assess the nighttime activity changes with administration of an NSAID, and to understand what demographic factors influenced nighttime activity.

## MATERIALS AND METHODS

### Design

Data for the current study was collected in two parts. Part A used previously unreported data from a masked, parallel, placebo-controlled clinical study evaluating dose titration of NSAIDs in dogs with naturally occurring OA pain which has been described already (*Wernham et al., 2011*). Part B used data gathered from a different cohort of dogs from a masked, placebo-controlled crossover study evaluating the effect of an NSAID versus placebo on nighttime activity in dogs with naturally occurring painful OA. The Institutional Animal Care and Use Committee (IACUC) approved both studies (IACUC #08-077-O, #07-188-O), and in all cases owners signed a written consent form following a detailed explanation of the study protocol.

### Study populations

Sixty and nineteen client-owned dogs with OA-associated pain and mobility impairment were included in Part A and Part B, respectively. Dogs of any breed, age, sex or weight were recruited. Group size in Part A was based on power calculations determined from the expected decrease in efficacy with NSAID dose titration, and published previously (*Wernham et al., 2011*). Study B was designed as a pilot study focused on evaluating nighttime restlessness. Power analysis for this was based on the known change in pain following NSAID administration, detected using validated owner assessment tools. The expected difference between placebo and active treatment was 30%, and the SD 40%, giving us an estimation of 16 dogs being required in the crossover study for an 80% power for owners detecting the improvement due to an NSAID.

#### *Inclusion criteria*

To be eligible for either study, dogs were required to have impaired mobility (according to their owners), and to be considered by the investigators as clinically appropriate candidates for pain relief using NSAIDs. They were required to have not received oral or parenteral steroids or injectable polysulphated glycosaminoglycans within the last 4 weeks, and owners were required to agree to stop administering NSAIDs 2 weeks prior to the start of the study. The screening evaluation included a physical, neurological and orthopedic examination and a complete blood count (CBC), serum biochemical analysis and urinalysis. Exclusion criteria included the presence of suspected or demonstrated systemic or local disease other than OA. Dogs were also excluded if they were suffering from recent joint instability such as cranial cruciate ligament rupture, or had undergone joint surgery within the previous 12 months. Dogs known to have intolerance to NSAIDs were excluded from the study. Other exclusion criteria were confirmed or suspected neurological, cardiac or endocrine disease. Results of all laboratory testing must have been either within the reference range values or considered clinically nonsignificant. If alanine aminotransferase or alkaline phosphatase were twice the high end of the reference range, then pre- and post- prandial bile acid tests were performed. If the postprandial bile acid value was within the reference, the dog was considered an acceptable candidate for the study. Digital radiographs of all clinically abnormal (painful) appendicular joints

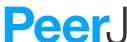

were taken under sedation.[1] Dogs with no detectable systemic disease, and with at least one appendicular joint where manipulation elicited an aversive response and whose radiographs showed the presence of OA, were included. Dogs were designated as being either predominantly 'forelimb' or 'hind limb' impaired. Owners were instructed not to change the management of dogs for the period of the studies.

## Study protocol

### Part A

As previously reported (*Wernham et al., 2011*), the study lasted for a total of 16 weeks. The present report concerns data collected over the nighttime periods of the first four weeks. On day 0, all dogs started the study but were not administered any analgesics for the subsequent 14 days. On day 14, all dogs received the full FDA-approved dose of NSAID (meloxicam, 0.2 mg/kg meloxicam orally on day 14, followed by 0.1 mg/kg orally, every evening, for the next 2 weeks). On day 0, the owners completed 3 clinical metrology instruments: the Client Specific Outcome Measures (CSOM) (*Lascelles et al., 2008a*), Helsinki Chronic Pain Index (HCPI) (*Hielm-Bjorkman, Rita & Tulamo, 2009*), and the Canine Brief Pain Inventory (CBPI) (*Brown et al., 2008*). Accelerometry data was collected continuously over the 4-week period using a collar-mounted accelerometer,[2] as previously described (*Wernham et al., 2011*). When the owner returned on day 14 and day 28 to collect/return the study medication, the accelerometer data were downloaded to a personal computer via a telemetric reader.

### Part B

This masked, placebo-controlled crossover study was conducted over a 5-week period. Inclusion criteria were as described above, and NSAID (meloxicam, dose as above) or placebo was administered for the first two weeks, followed by a 1 week wash-out period, followed by another two weeks of NSAID or placebo. Each dog received both NSAID and placebo in a random order. Randomization was based on a 3-block design, randomized in groups of 2 within each block. Randomization was blocked based on high, medium or low impairment, using the CSOM score (Low impairment group = scores of 1–14; intermediate = 15–29; high = 30–44). The placebo was visually identical to the regular meloxicam solution and prepared by the NCSU pharmacy.[3]

On day 0, owners completed 3 clinical metrology instruments: the CSOM, HCPI, and the CBPI. In addition, they completed the Sleep and Night Time Restlessness Evaluation Score (SNoRE) on days 14, 21, 28 and 35. Owners kept a daily diary of the times the household went to bed, and the time the household got up the following morning. These data were used to define the true nighttime period. Accelerometry data was collected as continuously over the 4-week period, as described above. On day 35, accelerometer data was downloaded to a personal computer as described above. Owners completed the clinical metrology instruments via a phone call on days 14, 21 and 28.

[1] Canon Medical CXDI-50G Sensor, Eklin Medical Systems, Santa Clara, CA.

[2] Actical Activity Monitor, Philips Respironics Co, Bend, OR.

[3] Methylcellulose (Ora Plus®) opacifier, and coloring (McCormick Food Colors Yellow and Blue).

## Outcome measures

Primary outcome measures, in both Part A and Part B, were activity monitor (accelerometer) counts over the nighttime periods. In addition, in Part B, the SNoRE was used as a subjective outcome measurement. The CSOM, HCPI, and the CBPI (total score, pain score and activity score) collected on day 0 were used to evaluate how the degree of impairment and pain (measured at the start of the study) influenced nighttime activity.

### Accelerometry (AM)

Spontaneous activity of dogs was measured with accelerometers as previously described (*Hansen et al., 2007*; *Wernham et al., 2011*). The epoch length was set to 1-minute. In Part A, data from the time period 12 am–5 am during 7 nights of the second week of the study (nights 7–13) and data from the time period 12 am–5 am during 7 nights of the fourth week of the study (nights 21–27) was used in the analysis. The period 12 am–5 am was chosen as these were the average times that a separate cohort of owners (Part B) went to bed and rose the following morning, less 1-hour at the start and the end. In Part B, accelerometer data from the individual nighttime periods for each day (as recorded by owners) were used, but the first hour of the nighttime, and the hour prior to waking were discarded in order to focus on the true nighttime period. This was performed in order to focus on the true nighttime period, and these periods varied from night to night. Only complete hours of accelerometer data were used.

### Sleep and Night Time Restlessness Evaluation Score (SNoRE)

SNoRE (Appendix S1) is a 6-item owner-administered clinical metrology instrument that was designed for this study and has not undergone any validation previously. The questionnaire was developed based on some of the most common observations of dog behaviors perceived as painful from the owners over a 10-year period in our clinical practice. It was designed to assess the quality of sleep of dogs over a 7-day time period.

## Statistical analysis

Data from Part A study was used to discover variables that were associated positively or negatively with nighttime activity, and to test for differences in nighttime activity between the baseline period (nights 7–13) and the treatment period (nights 21–27). Factors that were significant in Part A study were evaluated in Part B study for replication, and the effect of placebo versus NSAID on nighttime activity were also assessed in Part B.

To evaluate potential factors that were associated with nighttime activity, average activity per nighttime hour were calculated for each dog, and clinical and demographic variables were tested for association using appropriate tests of association. First, differences in activity between the baseline (nights 7–13) and NSAID treatment (nights 21–27) were tested using a repeated measures analysis of variance (ANOVA), with a term for time (by hour), and the baseline or treatment category. Based on negative results in this comparison, activity was averaged for each dog, and those averages were used in the subsequent analysis. Then, to test for differences according to demographic variables, repeated measures generalized linear models (linear regression) with appropriate degrees

of freedom were used to test for associations with variables and activity levels. Finally, to test for multivariable associations, forward step-wise variable selection using a Bayesian Information Criteria (BIC) selection criterion was used within the repeated measure regression to build a multivariable model and to automatically perform variable selection. BIC was chosen as the selection criteria given its model consistency properties (*Liang, Wang & Tsai, 2011*), and to control for concerns with over-fitting with the forward stepwise approach, a Bonferroni correction for the number of hypotheses tested was used for the multivariable analyses. The following variables were tested for univariate analysis and were entered as potential predictive variables in the stepwise modeling: Sex, Age, Weight, Predominantly Fore or Hind limbs affected, CSOM score at day 0, CBPI pain score at day 0, CBPI function score at day 0, CBPI total score at day 0, HCPI total score at day 0, Total pain score at day 0, and day of the week. To control for multiple comparisons in the regression modeling process, a Bonferroni correction was used to maintain a family-wise error rate of 0.05, and such that a $p$-values less than $0.05/10 = 0.005$ were considered statistically significant ($0.05/10$ was used as only 10 variables were found to be significant in univariate analysis, and only significant variables were used in the secondary analysis). Because the overall goals of the study were exploratory, the univariate analyses were considered at both nominal levels (not corrected for multiple comparisons) and using a Bonferroni correction, though we recognize that more conservative controls could be considered.

After variables were selected as significant in Part A study, data from Part B were used to replicate these associations (to help reduce any false positives from the multiple testing in Part A). Only significant variables were tested for association with activity using appropriate tests of hypothesis, as performed in Part A. Differences in activity between placebo and NSAID administration were tested using matched-pairs $t$-tests, and again, based on null results, the two activity scores for each hour per dog were averaged prior to subsequent analysis. Variables with $p$-values less than 0.05 were considered statistically significant.

SNoRE total values were reasonably normally distributed, and total scores from day 0 (baseline), day 14 and day 35 were compared (baseline and NSAID; baseline and placebo; placebo and NSAID) using a paired $t$-test, with a critical $p$-value of 0.016 to adjust for multiple comparisons. Additionally, the same procedure was repeated for each of the 6 individual questions on the SNoRE. The correlation between SNoRE values and CBPI pain scores measured at the three time points was calculated.

All analyses were performed in Stata v11 (www.stata.com) and R software (www.cran.org), and also JMP (JMP, SAS, Cary, NC, USA).

## RESULTS

In Part A, dogs were a mean of 9.29 years old (SD ± 2.83); 30.0 kg (SD ± 10.4) in bodyweight, and consisted of 11 spayed females and 8 neutered males. The mean total CBPI score (addition of all questions) of dogs in Part A was 46 (SD ± 18). In Part B, dogs were a mean of 9.96 years old (SD ± 2.42); 32.0 kg (SD ± 5.6) in bodyweight, and consisted of 11 spayed females and 8 neutered males. The mean total CBPI score (addition of all

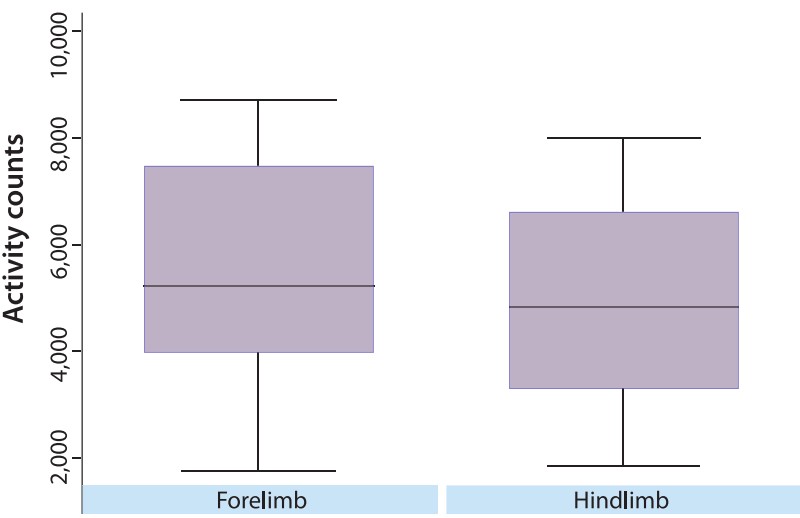

**Figure 1 Overall distribution of nighttime activity in dogs with predominately fore or hind limbs affected.** The overall distribution of activity counts in dogs with predominately fore or hind limb impairment. The $y$-axis represents mean activity counts (per minute). In both Part A and Part B, affected limb ($p < 0.001$ in both parts) was significantly associated with nighttime activity level.

questions) of dogs in Part B was 38 (SD $\pm$ 14). The degree of impairment (as measured by total CBPI scores) was not different between the groups ($P = 0.31$).

In Part A of the study there was a nominal increase in nighttime activity but no significant difference between nighttime activity during the baseline period (mean 4,642; SD $\pm$ 2,052) and the treatment period (mean 5,414; SD $\pm$ 2,020) ($p < 0.5976$, uncorrected), with baseline minus NSAID treatment mean hourly activity being $-772$ $\pm$ 2,931. There was also no significant association between treatment and activity in Part B ($p < 0.2645$, uncorrected). Values for the change in activity (per hour) calculated for placebo minus NSAID treatment (mean 121; SD $\pm$ 963) show a slight, but non-significant decrease in hourly nighttime activity in Part B.

The results of the variable selection in Part A indicated one variable (affected limb) that was statistically significant using a Bonferroni adjusted $p$-value, affected limb ($p < 0.001$, uncorrected). This variable was tested for association in Part B, and was also statistically significant ($p < 0.001$). In both studies, the average nighttime activity was higher for dogs with an affected forelimb than hind limb (Fig. 1). However, using data from Part A, comparison of CBPI scores indicated that the forelimb OA dogs ($n = 12$) had significantly lower total CBPI scores (less impaired) than the hindlimb OA dogs ($n = 48$) ($p = 0.014$). Additionally, the univariate analyses indicated some nominal associations (not statistically significant after multiple testing corrections) in both parts of the study. In Part A, weight ($p < 0.0013$), CBPI pain score ($p < 0.0301$), and total CSOM score ($p < 0.0302$) were nominally significant. In Part B, CBPI pain score ($p < 0.001$) and CSOM score ($p < 0.0374$) were nominally significant. Each of these variables was negatively correlated with activity, such that lower weight, lower CBPI pain score, and lower CSOM score were associated with higher activity. For weight, the correlation was $r = -0.44$

**Table 1  Table of change in SNoRE scores.** Mean change in scores between treatments and baseline for the SNoRE clinical metrology instrument. Negative changes indicate improvement in sleep. Critical $p$-value was adjusted to $p = 0.016$ within each question (or the total score) to reflect multiple comparisons within each question.

| | NSAID–Baseline | | Placebo–Baseline | | NSAID–Placebo | |
|---|---|---|---|---|---|---|
| | Mean difference | $p$-value | Mean difference | $p$-value | Mean difference | $p$-value |
| Question 1 | −1.05 | 0.0286 | −0.37 | 0.4209 | −3.58 | 0.2435 |
| Question 2 | −1.21 | **0.0149** | −0.26 | 0.5433 | −0.94 | 0.0462 |
| Question 3 | −1.37 | **0.0030** | −0.21 | 0.5311 | −1.15 | **0.0029** |
| Question 4 | −1.26 | 0.0175 | −0.21 | 0.7255 | −1.05 | 0.1036 |
| Question 5 | −0.89 | **0.0072** | −0.63 | 0.0688 | −0.26 | 0.3835 |
| Question 6 | 0.21 | 0.4647 | 0.42 | 0.3664 | −0.21 | 0.7112 |
| Total | −6.00 | **0.0012** | −1.26 | 0.4227 | −4.73 | 0.0496 |

(moderate) in Part A. For CBPI, $r = -0.24$ (modest) in Part A and $r = -0.42$ (moderate) in Part B. For CSOM, the correlation was $r = -0.41$ (moderate) for Part A, and $r = -0.38$ (moderate) in Part B.

The SNoRE instrument detected a positive improvement due to the NSAID ($p = 0.001$), and detected a difference between the NSAID and placebo ($p = 0.049$). Questions 2 and 3 appeared to be the best at detecting the positive effects of the NSAID on the quality of sleep (Table 1). There was a significant ($p < 0.001$) moderate ($r = 0.47$) correlation between the SNoRE and the CBPI pain score.

## DISCUSSION

Overall, we did not find evidence of nighttime restlessness, as measured by accelerometry, in dogs with naturally occurring OA. The SNoRE clinical metrology instrument did detect a subjective improvement in sleep with the provision of analgesia, indicating that quality of sleep is disturbed in naturally occurring OA in dogs due to pain. The SNoRE deserves further evaluation in a larger cohort of dogs, as do other measures of sleep disturbance.

One major criticism of this study is that we did not include an age-matched cohort of normal (non-OA) dogs to determine if nighttime activity differed between normal and OA dogs, and further work should evaluate this.

Accelerometry has been widely used in human studies as an objective assessment of sleep disturbances (*Sadeh, 2011*; *Wrzus et al., 2012*). However, relatively little work has been done in human OA patients. One study found that accelerometry did not distinguish between sleep efficiency in older persons with and without chronic pain, but sleep diaries did (*Lunde et al., 2010*). Interestingly, that study appears to reflect the results of our study where the quality of sleep was improved with NSAID treatment (pain alleviation), but accelerometry did not change. Another study suggested human patients with OA have higher levels of nocturnal body motility, but this was not proven (*Leigh et al., 1988*). Another study, that included accelerometry as a measure of nighttime restlessness, showed no effect of yoga on sleep disturbance when accelerometry data was evaluated, despite

the fact that participants reported significantly fewer nights with insomnia following an 8-week yoga program (*Taibi & Vitiello, 2011*). However, the study was relatively small (13 participants; accelerometry data available in 11), and was not masked. Regardless, collectively, it appears that sleep may be disturbed without any resulting measurable increase in movement as measured by simple actigraphy. Simultaneous use of multiple accelerometers and integration of the data to measure sleep position may be more useful (*Wrzus et al., 2012*), but as yet untried in OA patients. Indeed, overall the assumption that sleep disturbance from OA-pain is associated with increased movement has not been proven. In a rodent model of OA (iodoacetate), sleep patterns, as measured electrophysiologically, were altered (*Silva, Andersen & Tufik, 2008*; *Silva et al., 2011*); in adjuvant-induced arthritis, fragmented sleep patterns occurred following the induction of arthritis (*Landis, Robinson & Levine, 1988*); and in a model of gout arthritis, sleep-wake patterns were altered (*Guevara-Lopez et al., 2009*). However, none of these studies measured body movement or activity induced by arthritis in rodents. Further work is required to determine if accelerometry is an appropriate measure of sleep disturbance in any species.

In our study, the level of pain may not have been severe enough to elicit sleep disturbance. Although the severity of impairment in the present study reflects dogs that are recruited to OA-pain studies, further work should be directed at determining if a particular level of owner-noted impairment, or other phenotypic characteristics appear to be associated with the owner-noted sleep impairment. However, in a recent focus group study (*Woolhead et al., 2010*), night pain was found to occur amongst people with varying levels of current pain severity (*Woolhead et al., 2010*). Additionally, estimates of sleep disturbance in human OA patients suggest between 30 and 80% of OA patients suffer sleep disturbances (*Moldofsky, Lue & Saskin, 1987*; *Wilcox et al., 2000*; *Power, Perruccio & Badley, 2005*; *Woolhead et al., 2010*; *Taylor-Gjevre et al., 2011*), indicating a broad cross-section of the OA population can be affected. Despite 30 to 80% of OA human patients suffering sleep disturbances (*Moldofsky, Lue & Saskin, 1987*; *Power, Perruccio & Badley, 2005*; *Wilcox et al., 2000*; *Woolhead et al., 2010*; *Taylor-Gjevre et al., 2011*), a recent study showed no disturbances in slow-wave sleep in a rodent model of bilateral osteoarthritic pain (*Leys et al., 2013*), suggesting this naturally occurring model in dogs should be further evaluated for translational research utility.

Our results showed that dogs that were classified as having predominately forelimb impairment had significantly higher activity counts when compared to dogs with predominately hind limb impairment; however, the forelimb OA dogs were less impaired (as determined by owner scores) than the hindlimb OA dogs. Future studies should stratify for fore/hind limb involvement, or focus on one or the other phenotype.

The subjective clinical metrology instrument showed improvement in the quality of sleep related to the NSAID, compared to baseline, and over placebo. There was no difference between placebo and baseline. Despite this robust responsiveness validity, however, with no change in accelerometry the criterion validity of the instrument cannot be inferred. The improvement in scores following NSAID treatment in this masked,

placebo-controlled study, indicated responsiveness validity, suggest that this subjective instrument should be further tested and evaluated. It is unclear what aspects of sleep detected by the instrument were altered by the NSAID but, looking at the questions that appeared to be most responsive, we get some indication. The most responsive questions appeared to be questions asked about twitching and dreaming, followed by those asking about shifting position and vocalizing. The least-responsive questions appeared to be those asking about moving and pacing. When the responsiveness of the individual questions are viewed like this, the lack of effects on accelerometry-measured activity seem more inline with the SNoRE results. Further research could use objective video (*Hansen et al., 2007*; *Lascelles et al., 2008b*) and acoustic analysis of sleeping behavior to assess the effects of analgesic intervention. This would help further characterize the OA-dog as a model of human sleep disturbance, and would help understand if the improvements seen in the SNoRE reflected real changes in sleep quality. It was reassuring that the correlation between the SNoRE and the CBPI pain score (a validated measure of pain and pain relief in dogs with naturally occurring OA (*Brown et al., 2008*; *Walton et al., 2013*) was positive, significant and moderate. This suggests the SNoRE is measuring a similar construct, but likely different aspects. Future research should evaluate test-retest stability (reliability) of the SNoRE, and also evaluate criterion validity through the use of videography.

There are excellent induced large animal models of OA in dogs and horses (*Brandt, 2002*), and rodent models of chronic pain for the evaluation of sleep disturbance (*Leys et al., 2013*), and these have great utility. However, as previously suggested (*Mogil, 2009*) the use of such spontaneous disease preclinical models, where pain was not artificially induced and has a time course more similar to the human condition, and particularly where pain is spontaneous, could also be helpful in preclinical testing of analgesics and understanding the underlying mechanisms.

## ACKNOWLEDGEMENTS

The authors would like to acknowledge the assistance of Dr. Ben Wernham, Dr. Brian Trumpatori and Mr. Jon Hash in the collection of data, and Tonya Lee for manuscript editing assistance.

### Funding

The studies from which these data were taken were funded by Boehringer Ingelheim (Part A) and the Comparative Pain Research Laboratory (Part B). The funders had no role in study design, data collection and analysis, decision to publish, or preparation of the manuscript.

### Grant Disclosures

The following grant information was disclosed by the authors:
Boehringer Ingelheim (Part A).
Comparative Pain Research Laboratory (Part B).

## Competing Interests

BDXL has received honoraria for continuing education presentations from Boehringer Ingelheim, and has received research support from Boehringer Ingelheim. David Knazovicky, Andrea Tomas and Alison Motsinger-Reif report no conflicts of interest. B Duncan X. Lascelles is an Academic Editor for PeerJ.

## Author Contributions

- David Knazovicky and Andrea Tomas performed the experiments, wrote the paper, reviewed drafts of the paper.
- Alison Motsinger-Reif analyzed the data, contributed reagents/materials/analysis tools, wrote the paper, prepared figures and/or tables, reviewed drafts of the paper.
- B. Duncan X. Lascelles conceived and designed the experiments, performed the experiments, analyzed the data, contributed reagents/materials/analysis tools, wrote the paper, reviewed drafts of the paper.

## Animal Ethics

The following information was supplied relating to ethical approvals (i.e., approving body and any reference numbers):

The Institutional Animal Care and Use Committee (IACUC) approved both studies (IACUC #08-077-O, #07-188-O).

## Supplemental Information

Supplemental information for this article can be found online at http://dx.doi.org/10.7717/peerj.772#supplemental-information.

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
