# Peer review of "Initial evaluation of nighttime restlessness in a naturally occurring canine model of osteoarthritis pain"

_PeerJ, doi:10.7717/peerj.772_

## Round 0.1 · original submission · Major Revisions

Reviewer 2 has considerable experience using the same techniques as those employed in your manuscript, and raises a number of important issues to consider with respect to methods as well as the presentation of the Discussion section. For example, "What is more sensitive as outcome is the individual change in accelerometry related to the treatment each dog will receive (either placebo or meloxicam)." implies that you should augment your analyses to help ensure you are not missing an important effect. The Discussion section clearly needs more introspection on your findings. So I'm going to suggest that you resubmit the manuscript, and address each one of Reviewer 2's point individually.

Reviewer 1 ·

Basic reporting

Based on my evaluation of this study, I have no problems with their conservative conclusions or their methods.

Experimental design

The manuscript is well written and the experimental design is reasonable.

Validity of the findings

Although the results presented in the manuscript is minimal with mostly negative data, I think that it is a good new further insight into the relationship between osteoarthritic pain and sleep disturbance.

Annotated reviews are not available for download in order to protect the identity of reviewers who chose to remain anonymous.

·

Basic reporting

OK, however the lack of positive result led the authors for fishing to possible association, not so strongly supported by hypothesis. The main important critic relates to the Discussion that would need to be improved in clarity with a summary of results followed by an interpretation of results supported by literature data.

Experimental design

OK, however Line 66: The number of dogs used in Part B (n = 30) is in contradiction with the number mentioned in the abstract (n = 15) for the same part.

Validity of the findings

Introduction
Line 38: The abbreviation NSAID has not been defined earlier in the introduction.

Line 41-42: This statement is not supported by previous studies so it lacks credibility.

General question for the introduction: Why did you decide to choose NSAID such as meloxicam and why not another class of analgesic such as an opioid or another type of pain reliever other than the study you cited (Wernham et al., 2011)? I suppose it to be related to the retrospective nature of Part A, and its support by a Pharmaceutical Industry, but this would need to be explained, as the choice of NSAID is not without potential deleterious effect on sleep quality.

Material and methods
Line 123: This is a major bias of not having selected the same time periods for accelerometer evaluation during the night. It would have been more careful to discard in part A the same hour than in part B. Moreover, based in our large experience with accelerometry, we have largely described that a group effect in accelerometry would require a large sample size, with regards to the individual variability. What is more sensitive as outcome is the individual change in accelerometry related to the treatment each dog will receive (either placebo or meloxicam). This would largely simplify the statistical analysis and in consequence the clarity of the paper. Moreover, the count intensity is not always the best choice of outcome in accelerometry as we have reported in previous publication the association between accelerometry and CSOM (Rialland P et al. Clinical validity of outcome pain measures in naturally-occurring canine osteoarthritis. BMC Veterinary Research. 8:162, 10 Sep 2012. http://dx.doi.org/10.1186/1746-6148-8-162.) and most importantly the correlation between accelerometry duration of activity and kinetic outcome (Moreau M et al. A posteriori comparison of natural and surgical destabilization models of canine osteoarthritis. BioMed Research International. 2013:180453, Nov 2013.). Therefore such outcome could be considered to characterise sleepness quality.

Line 158-160: It is not clear the p-values used because Bonferonni correction indicated 10 possible comparisons (0.05/10) but 12 variables seems to be exploited in the stepwise modeling.

Results
Line 182: Is there an error of calculation? Baseline period (4642) minus NSAID treatment (5414) equal -772 and not -771.

Line 184-185: Given the lack of detail in the results of the placebo group and the treatment group, it is impossible to judge the repeatability of the results of part B to part A. However, results of part A are better detailed in the previous paragraph than part B.

Line 190-191: The figure 1 does not show if there is a statistical difference between forelimb and hind limb quite as in the text where it is absent too. This result lacks detailed information of the values and the statistical ‘’p’’ associated with these overall distribution of activity counts.

Line 197-198: Although the correlation is negative, the value of the correlation coefficient (r2) is far from -1 to be able to establish a possible association. The association seems to be a negatively moderate association. From what value of r2 do you consider that the association is respectable?

Line 200: There is nevertheless a strong tendency to observe a difference between the 2 groups (p=0.0049).

Table 1: Knowing that there is no difference between the comparison of the Placebo and Baseline group, how do you explain that the comparison of each of these 2 groups with the treatment group is so different? You never discuss these points in the Discussion section.

Discussion
The discussion alternates between paragraphs of previous studies in the literature and the summary of their results. The authors never discussed in detail the results obtained specifically in the figure 1 and the table 1 even if it is positive or negative results. On the other hand, they admit well the error to have not included a group of normal dog. It is obvious that their subjective and objective tools are lacking validity. It does not return clearly on the achievement of their purpose, objective and hypothesis of the research. Some structure of the Discussion is needed and conclusion is not clearly defined. The future perspectives are not either developed.

Additional comments

In this manuscript by Knazovicky and colleagues, the authors evaluated the use of objective accelerometry and subjective clinical metrology instruments to assess the nighttime activity changes with administration of a non-steroidal anti-inflammatory drug (NSAID) in a natural canine model of osteoarthritis (OA) pain. They also propose to identify the demographic factors that influenced nighttime activity. This is a pertinent goal despite the fact that in literature, studies on night pain is not a common subject and the result of all these studies seems to be controversial. The main structure of the article is respected (goal, objective and hypothesis are clearly identified in the introduction) but the results are not sufficiently detailed and this is important to enable reproducibility of the study. The criteria for inclusion of a dog is not enough specific, the degree of pain, quite as the level of disease affectation, are not well-defined and this could influence the analgesic treatment, as NSAID could be non sufficient to relieve pain. On the other hand, a key point of the study is the elaboration of the statistical tests used, what is often neglected. The Discussion reads more like a summary of the results interlaced with occasional literature review. It does not discuss what it is observed by theirs results. On the other hand, they admit their error to have not included a normal group of dog. This article demonstrated nothing in particular except suggesting a potential use of natural canine OA model to mimic the human pain-related nighttime sleep disturbance. The subjective and objective pain assessment tools used in this study failed to detect changes in nighttime activity when an NSAID is administrated. I don’t believe anything has actually been found here because of the lack of previous validity and repeatability of the tools used and some error in the experimental design. All the study is highly exploratory. They suggest only that the affected limb (forelimb or hind limb) could be a predictor of nighttime activity. Therefore, the authors could simplify their objectives and methodology (statistical analysis), and focus on the important discovery they brought with their work, even if negative, and highlight in particular the interest of such natural model of OA for its translationability to the human OA condition.

---

## Round 0.2 · accepted · Accept

I am recommending acceptance of this manuscript for publication at this time. Although it is clear that there are different interpretations of this study by the authors and a reviewer, I believe that airing these differences through the reviewer comments and author rebuttal will be valuable for interested readers. It will also hopefully motivate additional research to resolve the extant differences and questions that remain.